# A Method for Predicting Inertial Navigation System Positioning Errors Using a Back Propagation Neural Network Based on a Particle Swarm Optimization Algorithm

**DOI:** 10.3390/s24123722

**Published:** 2024-06-07

**Authors:** Yabo Wang, Ruihan Jiao, Tingxiao Wei, Zhaoxing Guo, Yueyang Ben

**Affiliations:** 1Wuhan Second Ship Research and Design Institute, Wuhan 430205, China; 18007195551m@sina.cn; 2School of Intelligent Science and Engineer, Harbin Engineering University, Harbin 150001, China; wtx15776482665@163.com (T.W.); gzx13946003678@outloook.com (Z.G.); byy@hrbeu.edu.cn (Y.B.)

**Keywords:** GPS denial, GPS/SINS integrated navigation system, backpropagation neural network, particle swarm optimization

## Abstract

In order to reduce the position errors of the Global Positioning System/Strapdown Inertial Navigation System (GPS/SINS) integrated navigation system during GPS denial, this paper proposes a method based on the Particle Swarm Optimization–Back Propagation Neural Network (PSO-BPNN) to replace the GPS for positioning. The model relates the position information, velocity information, attitude information output by the SINS, and the navigation time to the position errors between the position information output by the SINS and the actual position information. The performance of the model is compared with the BPNN through an actual ship experiment. The results show that the PSO-BPNN can obviously reduce the position errors in the case of GPS signal denial.

## 1. Introduction

In the integrated navigation system, the primary navigation system operates continuously. Concurrently, other navigation systems provide correction data to enhance the positioning accuracy of the system and continuously supply navigation parameters [1,2,3,4]. The most common combination for integrated navigation systems is the employment of SINS as the primary navigation system, with the GPS serving as the auxiliary navigation system. When the GPS is denied, the positioning errors of the SINS will diverge rapidly over time. Currently, the introduction of the BPNN as a substitution for the GPS is a conventional approach to assist in the prediction and compensation of SINS position errors during GPS denial. The ability of the BPNN to handle complex nonlinear relationships enables the precise fitting of the relationship between the SINS position errors and variables associated with them. This helps with accurate prediction and the compensation of the SINS positioning errors.

Artificial neural networks have been widely applied in practical engineering. N. Shaukat et al. [5] proposed enhancing Radial Basis Function Neural Networks (RBFNN) to compensate for integrated navigation systems. Considering the variable time delays between GPS measurement intervals, a self-regressive neural network fusion architecture was introduced in [6]. Reference [7] proposed a method for the error feedback correction of Micro-Electro-Mechanical Systems—Strapdown Inertial Navigation System (MEMS-SINS) based on neural network prediction during GPS denial. It employed an RBF Neural Network, utilizing the outputs of the MEMS-SINS gyroscopes and accelerometers as inputs and the position, velocity, and attitude errors of the MEMS-SINS as outputs. The method is validated through ground vehicle tests, demonstrating that the MEMS-SINS can achieve high-precision navigation information through the neural network, enabling continuous navigation. In reference [8], to address the issue of the filter accuracy divergence caused by system uncertainty errors, a novel algorithm was introduced that uses a Model Predictive Filter (MPF) to predict and correct the error propagation in a feedforward neural network. This novel feedforward algorithm fits and learns the nonlinear relationships of the system, and through simulation data, it was verified that its application effects in integrated navigation were significantly superior to those of the Extended Kalman Filter (EKF) algorithm. In the study presented in [9], a Multilayer Perceptron (MLP) Neural Networks was suggested to provide pseudo-GPS positions for compensating INS errors in the case of GPS denial. To address the limitations of GPS and WIFI signals in complex indoor environments, the study in [10] proposed an indoor positioning technology that combined the real-time capabilities of inertial navigation with the high precision of machine learning. The validation demonstrated that this method can significantly reduce the positioning time while ensuring the accuracy of the location determination. In [11], a fuzzy neural network topology for an integrated GPS/INS/odometer system was constructed. Additionally, the work made predictions for integrated navigation sequences using adaptive fuzzy neurons, fuzzy inference systems, Recurrent Neural Networks (RNNs), and Wavelet Neural Networks. Reference [12] employed a multilayer BPNN instead of Kalman filtering for transfer alignment in inertial navigation systems, achieving comparable accuracy to Kalman filtering while effectively reducing the computational time and improving the real-time capability of the system. The BPNN is likely to become entangled in local optima, thereby failing to attain global optimal solutions. To address this issue, the authors of [13] proposed a novel fusion algorithm containing additional momentum and hidden unit competition to optimize the BPNN. It can efficiently predict the increment in the position and compensate for the INS error accumulation during a GPS outage. Southeast University has proposed an improved BPNN adaptive algorithm that integrates a Genetic Algorithm (GA) and has also optimized the training process of the GA [14]. In [15], the GA-BPNN was utilized to predict a ship’s trajectory and showed a higher prediction accuracy than traditional BPNN. In addition to the GA, there are other types of artificial intelligence methods, such as ensemble models [16] and Support Vector Machine (SVM) [17], which have been employed to overcome the propensity of neural networks to become entangled in local optima. However, these algorithms often need complex feature engineering and exhibit low computational efficiency. To overcome the drawbacks of the BPNN in handling long-sequence data and retaining information over extended periods, scholars have employed the Long Short-Term Memory (LSTM) Neural Network in handling inertial navigation problems. Huazhong University of Science and Technology [18] proposed a filtering technique for gyroscope random errors based on the LSTM Neural Network. By utilizing this model, the standard deviation of the filtered gyroscope random errors was reduced compared to the original data, thereby enhancing the measurement precision of the gyroscope. Y.X. Zhang [19] proposed an integration method based on the LSTM Neural Network, which can adaptively utilize historical data without the need for additional inputs.

As the output of a BPNN is entirely determined by the input at the current moment, this network is characterized by a more rapid response to anomalous changes in the vehicle’s trajectory when compared to RNNs such as an LSTM Neural Network in navigation problems. An INS exhibits complex strong nonlinear characteristics, and its navigation parameters show a high dimensionality. The BPNN demonstrates a strong capability in fitting complex nonlinear relationships and has the advantages of straightforward principles. Synthesizing these attributes, this paper intends to employ the BPNN to assist the integrated navigation system. According to the theory proposed in article [13], the characteristics of the INS errors include their variability during the training and predicting process, and based on the method, this work also adds navigation time into the input of the neural network. However, the BPNN is likely to become entangled in local optima, which can lead to significant prediction errors. Although many optimization algorithms were proposed in the article mentioned above, they were exclusively utilized for the optimization of the BPNN. Moreover, the GA algorithm exhibits a long computation time and a low searching efficiency, which cannot meet the real-time requirements of the inertial navigation system. The PSO algorithm can easily converge to the global optimum, and its convergence rate is faster than that of the GA. It also does well in identifying optimal values in high-dimensional data spaces. Considering the aforementioned advantages of the PSO and its ability to effectively satisfy the characteristics and practical needs of the inertial navigation systems during actual usage, this work employed the PSO algorithm to optimize the BPNN. By utilizing the weights obtained during each training iteration of the BPNN, the PSO algorithm was applied to identify the optimal weights. This process enables the BPNN to escape from the local optima, thereby enhancing the model’s generalization capability and subsequently optimizing the performance of the BPNN. Based on actual ship data, the study conducted comparative validation experiments between the PSO-BPNN model and the traditional BPNN model.

This paper addresses the diverging positioning accuracy of a SINS in integrated navigation systems during GPS denial, by optimizing the BPNN through the PSO algorithm. This approach aims to solve the application problems of integrated navigation systems in special environments.

The structural arrangement of this paper is as follows:

Section 1 elucidates the significance of the research presented in this paper. Section 2 exposits the error model of the SINS. Section 3 introduces the proposed BPNN, as well as the PSO algorithm designed for optimizing the BPNN. Section 4 details the environment and outcomes of the actual ship experiments. Section 5 offers a summary of the experimental results.

## 2. Error Equations of Strapdown Inertial Navigation [20]

This section presents the error equations of the SINS, explaining the propagation process of position errors.

### 2.1. Attitude Error Equation

The attitude error equation is as follow:(1)$${\dot{\Phi }}^{n}=\Phi^{n}\times \left(\omega_{ie}^{n}+\omega_{en}^{n}\right)+\delta \omega_{ie}^{n}+\delta \omega_{en}^{n}-\varepsilon^{n}$$
where $\omega_{ie}^{n}$ is the rotational angular velocity of the navigation coordinate system relative to the Earth in the n frame, $\omega_{en}^{n}$ is the rotational angular velocity of the Earth relative to the inertial space in the n frame, $\delta \omega_{ie}^{n}$ and $\delta \omega_{en}^{n}$ correspond to the computational errors for $\omega_{ie}^{n}$ and $\omega_{en}^{n}$, $\Phi^{n}$ represents the attitude error angle in the n frame, and $\varepsilon^{n}$ is the gyro drift in the n frame.

### 2.2. Velocity Error Equation

The velocity error equation is as follow:(2)$$\delta {\dot{V}}_{e}^{n}=f^{n}\times \Phi^{n}-\left(2\delta \omega_{ie}^{n}+\delta \omega_{en}^{n}\right)\times V_{e}^{n}-\left(2\omega_{ie}^{n}+\omega_{en}^{n}\right)\times \delta V_{e}^{n}+\delta g^{n}+\nabla^{n}$$
where $f^{n}$ represents the measured values of the accelerometer in the n frame, $V_{e}^{n}$ is the velocity of the vehicle relative to the Earth in the n frame, $\delta V_{e}^{n}$ is the computational error of the $V_{e}^{n}$, $\delta g^{n}$ is the error in gravitational acceleration, and $\nabla^{n}$ is the zero-offset error of the accelerometer in the n frame.

### 2.3. Position Error Equation

By applying the total differentiation on both sides of the position computation equation, the position error equation can be expressed as follows:(3)$$\left\{ \begin{array}{l} \delta \dot{L}=\frac{\delta V_{y}}{R_{M}}\\ \delta \dot{\lambda }=\frac{\delta V_{x}}{R_{N}cosL}+\frac{V_{x}}{R_{N}cosL}tanL\delta L \end{array}\right.$$
where L is the latitude, λ is the longitude, $R_{N}$ and $R_{M}$ correspond to the meridian radius of curvature and the prime vertical radius of curvature, and $V_{x}$ and $V_{y}$ represent the eastward and northward velocities.

### 2.4. Kalman Filtering

This work applied a loose integration model to the GPS and SINS. In this model, the Kalman filter receives the different output values of the same navigation parameter from the two navigation systems. After filtering calculations, it estimates the error quantities. The estimated values of the SINS’s errors are then used to correct the parameters output by the SINS, thereby obtaining the optimal estimation of the navigation parameters. Figure 1 shows the working process of the Kalman filter.

The Kalman filter employs a recursive approach for a priori calculation of state parameters, utilizing the state estimation value and the mean square error matrix of the state estimation from the previous moment to predict the state parameters. It combines the current moment’s observation values to calculate the state estimation value and the mean square error matrix of the state estimation for the current moment. The process follows a “prediction–correction” formulation for recursion, thereby enabling real-time estimation of the system’s state parameters.

The state equation and measurement equation of the Kalman filter are as follows:(4)$$\left\{ \begin{array}{l} X_{k}=\Phi_{k,k-1}X_{k-1}+\omega_{k-1}\\ z_{k}=X_{k}+v_{k} \end{array}\right.$$
where $X_{k}$ and $z_{k}$ are the state vector and measurement vector at time instance k; $\Phi_{k,k-1}$ is the state transition matrix, and $H_{k}$ is referred to the measurement matrix; $\omega_{k-1}$ and $v_{k}$ are the process noise vector and measurement noise vector. They are uncorrelated error vectors, and both are sequences of zero-mean Gaussian white noise vectors.

The more general expressions for the state one-step prediction and the state one-step prediction mean squared error matrix of the state estimation that can be obtained through derivation are as follows:(5)$$\left\{ \begin{array}{l} {\hat{X}}_{k,k-1}=\Phi_{k,k-1}{\hat{X}}_{k-1}\\ P_{k,k-1}=\Phi_{k,k-1}P_{k-1}\Phi_{k,k-1}^{T}+Q_{k-1} \end{array}\right.$$
where $Q_{k-1}$ is the variance matrix of the system noise, which is required to be a known positive semi-definite matrix in the Kalman filter. Based on the recursive least squares estimation method, the current moment’s state estimation and the mean square error matrix of the state estimation can be derived as follows:(6)$$\left\{ \begin{array}{l} K_{k}=P_{k,k-1}H_{k}^{T}{\left(H_{k}P_{k,k-1}H_{k}^{T}+R_{k}\right)}^{-1}\\ {\hat{X}}_{k}={\hat{X}}_{k,k-1}+K_{k}\left(z_{k}-H_{k}{\hat{X}}_{k,k-1}\right)\\ P_{k}=\left(I-K_{k}H_{k}\right)P_{k,k-1}{\left(I-K_{k}H_{k}\right)}^{T}+K_{k}R_{k}K_{k}^{T} \end{array}\right.$$
where $R_{k}$ represents the variance matrix of the measurement noise, which is required to be a known positive definite matrix in the Kalman filter.

In (6), the first equation is the Kalman gain equation, and the matrix derived from it is the Kalman gain matrix. It is determined under the constraint criterion of minimizing the state variance and dictates the weight of the current estimation between one-step prediction and measurement. The second equation is the state estimation calculation equation, and the third equation is the estimation error covariance equation.

## 3. Optimization Method for the Backpropagation Neural Network

### 3.1. Principle of the Backpropagation Neural Network

The name of the Backpropagation Neural Network comes from the “backpropagation of error” process within its algorithm. Its structure can be divided into three components: the input layer, the hidden layer, and the output layer. Each layer consists of multiple neurons. The propagation process involves each neuron receiving inputs from the preceding layer of neurons, multiplying these inputs by their corresponding weights, summing them with the bias, and then undergoing a nonlinear transformation through the neuron’s activation function. The transformed result is subsequently passed on to the next layer of neurons.

The Backpropagation Neural Network employs a gradient descent method for self-learning and updating. When a set of desired outputs and input data are fed into the BPNN, the response values of the “neurons” propagate from the input layer through the hidden layer to the output layer. After being processed by the activation function, the actual output is obtained at the output layer. By comparing the difference between the actual output and the desired output, with the goal of minimizing this discrepancy, the weights and biases between neurons are updated layer by layer, starting from the output layer to the hidden layer and then to the input layer. This process of “forward computation of output–backward propagation of error” continues iteratively until the error is corrected to within a preset allowable error range, at which point the self-learning phase of the BPNN concludes. The Figure 2 shows the structure of a common BPNN.

Define $W_{ij}^{\left(l\right)}$ as the weight between the j-th neuron of l−1-th layer and the i-th neuron of l-th layer, and let $b_{i}^{\left(l\right)}$ represent the bias of the i-th neuron in the output layer. The process can be expressed as follows:(7)$$\begin{array}{c} net_{i}^{\left(l\right)}=\sum_{j=1}^{M_{l-1}}W_{ij}^{\left(l\right)}h_{j}^{\left(l-1\right)}+b_{i}^{\left(l\right)}\\ h_{j}^{\left(l\right)}=f\left(net_{i}^{\left(l\right)}\right) \end{array}$$
where $net_{i}^{\left(l\right)}$ is the input to i-th neuron in the output layer, f(•) is the activation function, $h_{j}^{\left(l-1\right)}$ is the output of the j-th neuron of l−1-th layer, and $M_{l-1}$ is the number of neurons of l−1-th layer.

Define the error function as E, with the minimum error ε and the maximum number of iterations set as M. Suppose there exists an m-dimensional input vector x={x(1),…x(m)}; let d(p) represent the network’s expected output for the p-th set of input data and y(p) represent the actual output of the network for the p-th set of input. Let n represent the number of neurons in the output layer. The symbol k represents the k-th neuron. The error function can be expressed as
(8)$$E=\frac{1}{2m}\sum_{p=1}^{m}\sum_{k=1}^{n}{\left(d_{k}\left(p\right)-y_{k}\left(p\right)\right)}^{2}$$

The partial derivative of the error function with the weights and biases for the p-th data set at the l-th layer can be expressed as follows:(9)$$\begin{array}{c} \frac{\partial E\left(p\right)}{\partial W_{ij}^{\left(l\right)}}=\delta_{i}^{\left(l\right)}h_{j}^{l-1}\\ \frac{\partial E\left(p\right)}{\partial b_{i}^{\left(l\right)}}=\delta_{i}^{\left(l\right)} \end{array}$$

$\delta_{i}^{\left(l\right)}$ can be expressed as follows:(10)$$\begin{array}{c} \delta_{i}^{\left(l\right)}={\sum_{k=1}^{M_{l+1}}{W_{ki}}^{\left(l+1\right)}\delta_{k}^{\left(l+1\right)}f{\left(x\right)}^{\prime }|}_{x=net_{i}^{\left(l\right)}}\\ \delta_{k}^{\left(l+1\right)}={-\left(d_{k}\left(p\right)-y_{k}\left(p\right)\right)f{\left(x\right)}^{\prime }|}_{x=net_{i}^{\left(l+1\right)}} \end{array}$$

Upon obtaining the partial derivatives, the weights and biases can be calculated. Subsequently, the error function E is determined and compared with the minimum error to assess whether the error function has fallen below it. The learning process concludes once the error meets the specified condition or the number of training iterations reaches the maximum number of iterations. If the criteria are not met, the learning continues.

### 3.2. The Backpropagation Neural Network Based on Particle Swarm Optimization Algorithm

The propensity of the BPNN to become entangled in local optima, coupled with the potential for larger errors, leads to less ideal prediction performance. Utilizing the PSO algorithm to optimize the BPNN, this method treats the weights generated during the neural network training process as particles for training. The optimal weights are identified when the maximum number of iterations is reached or the error reduces to a certain level. Apart from the strengths of the BPNN, the optimized network model overcomes the propensity of the BPNN to become entangled in local optima and the generation of larger errors. Consequently, this enhancement leads to an improvement in the prediction performance of the BPNN.

#### 3.2.1. The Principle of the Particle Swarm Optimization Algorithm

Particle Swarm Optimization is a swarm intelligence optimization algorithm. In this algorithm, each particle represents a potential solution to the problem. The fitness function is used to calculate the adaptability of each particle. Based on the calculation results, the velocity and position of the particles can be updated. The velocity of the particles dictates their movement direction and distance, thereby achieving the optimization of individuals in the solution space.

Suppose a D-dimensional space contains a swarm of n particles denoted as $X=(X_{1},X_{2},\cdots ,X_{n})$. Let $X_{i}={\left[x_{i1},x_{i2},\cdots ,x_{iD}\right]}^{T}$ represent the position of the i-th particle, $V_{i}={\left[v_{i1},v_{i2},\cdots ,v_{iD}\right]}^{T}$ represent its velocity, $P_{i}={\left[P_{i1},P_{i2},\cdots ,P_{iD}\right]}^{T}$ represent the individual best values achieved by the i-th particle, and $P_{g}={\left[P_{g1},P_{g2},\cdots ,P_{gD}\right]}^{T}$ represent the global best values obtained by the entire swarm. The steps of the PSO algorithm are listed in Algorithm 1.
**Algorithm 1** Steps of the PSO algorithm**Step 1**: Initialize the position and velocity vectors of the particle swarm.**Step 2**: Calculate the fitness value corresponding to each particle’s position based on the objective function, and evaluate the quality of the solutions based on the magnitude of the fitness values.**Step 3:** Identify the individual extreme values and the global extreme values of the particle swarm based on the fitness values.**Step 4**: Update the velocity and position of the particle swarm using the following formulas:
(11)$$V_{id}^{k+1}=\omega V_{id}^{k}+c_{1}r_{1}\left(P_{id}^{k}-X_{id}^{k}\right)+c_{2}r_{2}\left(P_{gd}^{k}-X_{id}^{k}\right)$$(12)$$X_{id}^{k+1}=X_{id}^{k}+V_{id}^{k+1}$$
where ω represents the inertia weight, d=1,2,⋯,D, i=1,2,⋯,n; k represents the current iteration number; $V_{id}$ represents the velocity of the particle; $c_{1}$ and $c_{2}$ are non-negative constants, referred to as acceleration factors; $r_{1}$ and $r_{2}$ are random numbers distributed in the range [0,1]. **Step 5**: Determine whether the termination criteria are met. If the criteria are satisfied, output the global optimum and conclude the algorithm; otherwise, revert to step 2 and continue with further iterations.

#### 3.2.2. PSO-Based Optimization Algorithm for Backpropagation Neural Network

The PSO algorithm can optimize the weights of the BPNN, thereby enhancing the network’s nonlinear fitting capability. The method of the PSO-BP algorithm [21] is listed in Algorithm 2.
**Algorithm 2** Steps of the PSO-BP algorithm**Step 1**: Transform the initial weights of the BPNN into particles within the framework of the PSO algorithm, with random initialization of the particles’ velocities and positions.**Step 2**: Establish the number of training set samples, test set samples, hidden layer nodes, particle swarm size, iteration count, inertia weight, and acceleration factors for the network.**Step 3**: Calculate the fitness values for each particle.**Step 4**: When the conditions for fitness value updating are satisfied, update the positions and velocities of each particle according to (11) and (12), and record the best positions of each particle.**Step 5**: Record the global best position.**Step 6**: Assess whether the termination criteria are satisfied; if so, output the global optima, obtain the optimal weights, and conclude the algorithm; otherwise, reinitialize the velocities and positions of the particles and reiterate the process.

### 3.3. Error Prediction and Compensation Methods for Integrated Navigation Systems

The PSO-BPNN proposed in this paper takes the position information, velocity information, attitude information output by the SINS and the navigation time as inputs. The position errors between the position information output by the SINS and the actual position information is used as the output.

When GPS is available, the PSO-BPNN is trained. The training algorithm iteratively adjusts the parameters of each layer to minimize the mean square error, thereby achieving the optimal neural network model. During GPS denial, still employing the position information, velocity information, attitude information output by the SINS and the navigation time as inputs to the already trained PSO-BPNN, one can predict the position errors of the SINS. Figure 3 shows the training and predicting process of the method proposed in this article.

## 4. Experimental Validation on an Actual Ship and Result Analysis

### 4.1. The Design of the Actual Ship Experiment

This work employed actual ship data to conduct a validation experiment. The experimental conditions were as follows:

The experimental ship was outfitted with a SINS/GPS integrated navigation system constructed from a self-developed fiber-optic SINS and GPS. Concurrently, a fiber-optic SINS known as PHINS, developed by the French company iXblue (Saint-Germain en Laye, France), served as the benchmark. When PHINS is in the integrated navigation mode, it is capable of providing high-precision attitude information (with an error not exceeding 0.01°). The performance metrics of the self-developed fiber-optic Strapdown Inertial Navigation System are characterized by a gyroscopic constant drift rate of approximately 0.01°/h, an accelerometer zero bias of about 50 μg, and an inertial sampler frequency of 100 Hz.

The experiment utilized 14 h of actual ship data, with 10 h of data allocated for training and 4 h designated for prediction. The performance of two types of neural networks in assisting the prediction of the SINS positioning errors during GPS denial was compared: the BPNN and the PSO-BPNN. Each model employed the same depth and parameters to ensure a fair comparison. The metrics for comparison included the latitude error and the longitude error.

### 4.2. Results and Analysis of the Actual Ship Experiment

To quantitatively compare the performance of the two methods, this work selected the Absolute Mean (*AM*) and the Root Mean Square Error (*RMSE*) as evaluation metrics.
(13)$$\left\{ \begin{array}{l} AM=\frac{\left(\left|x_{1}\right|+\left|x_{2}\right|+\cdots +\left|x_{n}\right|\right)}{n}\\ RMSE=\sqrt{\frac{1}{n}\sum_{i=1}^{n}{\left(y_{i}-{\hat{y}}_{i}\right)}^{2}} \end{array}\right.$$
where $\left|x_{i}\right|$ represents the absolute value of the i-th data point, n is the number of data points in the dataset, ${\hat{y}}_{i}$ is the expected output of the model, and $y_{i}$ is the actual output of the model.

The loss function curves of the two methods are shown in Figure 4. The positioning error curves of the SINS are shown in Figure 5. They are the difference between the position information output by the SINS and the actual position data. The positioning error curves for both methods are shown in Figure 6. They are the difference between the predicting position errors and the actual position errors.

Through the comparation of the loss function curves, it is evident that the loss function curve of the PSO-BPNN descends more rapidly than that of the BPNN. This indicates that the BPNN model optimized by the PSO algorithm exhibits a faster learning rate and higher learning efficiency. Ultimately, it exhibits a smaller mean square error, which indicates that the PSO has successfully optimized the BPNN by identifying the optimal weights for the neural network model.

Analysis of Figure 5 indicates that during periods of GPS denial, the positioning errors of the SINS gradually diverge over time, and Table 1 shows that the AM of the SINS longitude error is 4070.6 m, and the latitude error is 11,950 m. The error is too excessive to locate precisely, which fails to satisfy the requirements for navigation accuracy.

According to the Figure 6 and Table 2, the AM of the longitude error for the BPNN is 7.9319 m, the RMSE is 9.7788 m, the maximum value is 45.03 m, and the minimum value is −42.06 m; for the latitude error, the AM is 11.4662 m, the RMSE is 14.6223 m, the maximum value is 70.84 m, and the minimum value is −77.69 m. It is evident that the BPNN is capable of suppressing the divergence of errors over time, suggesting that it can serve as a substitute for a GPS in assisting the integrated navigation system during GPS signal denial.

The AM of the longitude error for the PSO-BPNN is 3.3339 m, the RMSE is 4.3494 m, the maximum value is 23.29 m, and the minimum value is −25.35 m; for the latitude error, the AM is 3.4728 m, the RMSE is 4.3309 m, the maximum value is 16.85 m, and the minimum value is −16.16 m. Compared to the BPNN, the PSO-BPNN achieves a reduction of 57.97% in the AM of longitude error, 55.52% in the RMSE of longitude error, 69.71% in the AM of latitude error, and 70.38% in the RMSE of latitude error, and the fluctuation range of the data has also been reduced. These demonstrate that the predictive performance of the PSO-BPNN is superior to that of the BPNN. It is also capable of suppressing the divergence of errors over time and can confine the errors to a small range of fluctuation.

## 5. Conclusions

This paper proposes a method for predicting the positioning errors of SINS based on a Particle Swarm Optimization–Back Propagation Neural Network model. When the GPS signal is available, the output information of the SINS, navigation time, and the position errors between the position information output by the SINS and the actual position information are collected and trained by the PSO-BPNN module. During GPS denial, the well-trained model will provide the position errors. Through comparative experiments, the model proposed in this paper showed a higher accuracy on predicting the position over the traditional BPNN model. It provides an effective way to solve the application challenges of integrated navigation in specialized environments. In the future study, the state of the navigation system, such as the motion status will be considered to affect the predictive performance of the model. More experiments will be conducted based on this foundation.

## Figures and Tables

**Figure 1 sensors-24-03722-f001:**
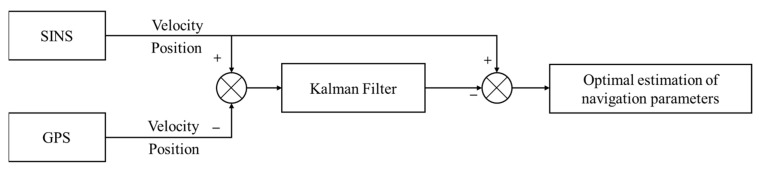
GPS/SINS pine integrated navigation system model.

**Figure 2 sensors-24-03722-f002:**
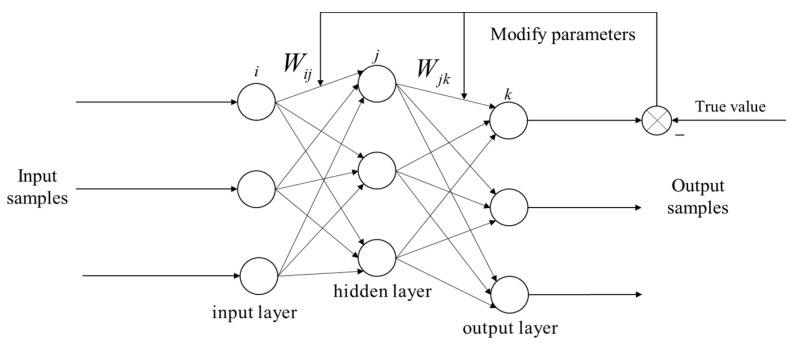
The three-layer backpropagation neural network structure.

**Figure 3 sensors-24-03722-f003:**
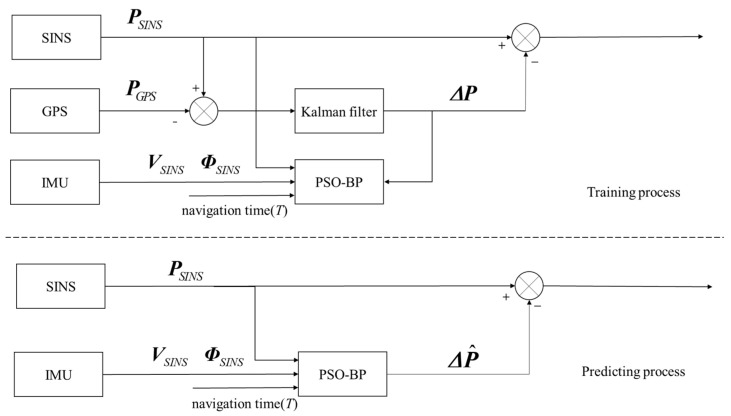
The training and prediction process of the PSO-BP neural network.

**Figure 4 sensors-24-03722-f004:**
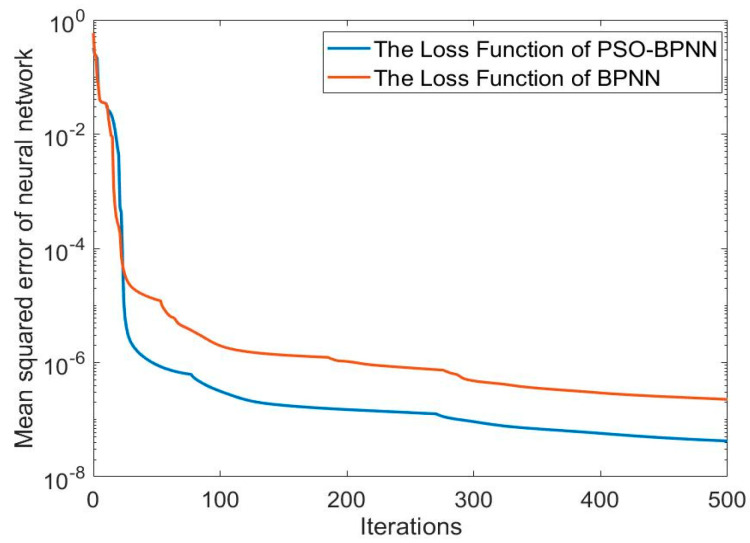
The loss function curves of the two methods.

**Figure 5 sensors-24-03722-f005:**
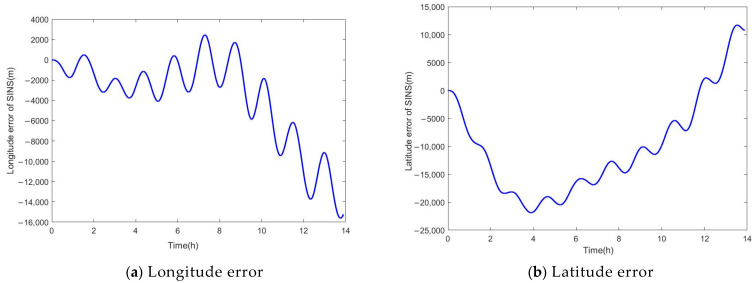
The position error curves of the SINS.

**Figure 6 sensors-24-03722-f006:**
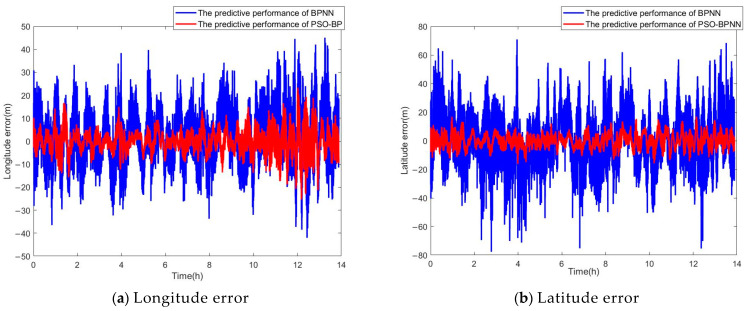
The position error curves of the two methods.

**Table 1 sensors-24-03722-t001:** The statistics of the SINS positioning errors.

Error	AM	RMSE
Longitude error (m)	4070.6	30.459
Latitude error (m)	11,950	21.540

**Table 2 sensors-24-03722-t002:** Analysis of the experimental results.

Method	BPNN				PSO-BPNN			
Error Index	AM	RMSE	MAX	MIN	AM	RMSE	MAX	MIN
Longitude error (m)	7.9319	9.7788	45.03	−42.06	3.3339	4.3494	23.29	−25.35
Latitude error (m)	11.4662	14.6223	70.84	−77.69	3.4728	4.3309	16.85	−16.16

## Data Availability

The data are contained within the article.

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
