# Peer review of "A Method for Predicting Inertial Navigation System Positioning Errors Using a Back Propagation Neural Network Based on a Particle Swarm Optimization Algorithm"

_sensors, 2024, doi:10.3390/s24123722_

Round 1

Reviewer 1 Report

Comments and Suggestions for Authors

This study proposed the integrated navigation system method based on the Particle Swarm Optimization-Back Propagation Neural Network (PSO-BPNN), which utilizes neural networks to replace the GPS for positioning. It is interesting and helpful for the research of predicting intertial navigation system positioning.

However, there are some issues or questions should be addressed:

1) In the abstract, the important objectiveness and aiming problems should be explained clearly. The comparative methods and important findings, such as other intertial navigtion system positioning error reduction methods, should also be described clearly. 

2) In the Section 1 of Introduction, the objectiveness and the novel meaning of this study  should be clearly explained. The research gaps of this study, especially the problems in previous studies about the inertial navigation system positioning errors, should be clearly explained and listed. 

3) In the Section 2 of Error equations of strapdown inertial navigation, there should be a framework flowchart of this study to indicate the novelty and the whole processing. It is not clear if the methods or equations in this section are proposed by the authors, as it includes more commonly used processing algorithms.

4) In the Section 3 of Optimization method for backpropagation neural network, it should be explained the relationships between the strapdown inertial navigation in Section 2 and the optimization method for BPNN in this Section. How to apply the optimization method for backpropagation neural network in INS error reduction, such as the input data and the network architecture?

5) In the Section 4 of Experimental validation on actual ship and result analysis, there are should some description of these experiments and their meaning. The authors should check the calculation for AM in equation (13), as absolute mean value or error? Why are the position errors in Figure 5 and Table 3 much larger than ones in Figure 6 and Table 4? The reasons should be explained clearly. The advantages and disadvantages of this study should also be explained clearly. 

6) There were errors in writing in the content, such as spaces between sentences; a period at the end of a sentence. The important findings and future research fields need to be highlighted briefly in Section 5.

7) Some related studies should be included and cited in this study, such as: Wang, et al, 2019, doi: https://doi.org/10.3390/rs11131540; et al.

Comments on the Quality of English Language

Minor editing of English language is needed.

Reviewer 2 Report

Comments and Suggestions for Authors

1.     The introduction is not clear. The main idea of the second paragraph is "Artificial neural networks have been widely applied in practical engineering." For this reason, the author explains a large amount of literature that is not closely related to this study, which does not help readers to understand the work of this paper. So, it is recommended to delete it. The third paragraph mentions the advantages of BPNN, and the algorithm of this paper is based on BPNN, so it is necessary to collect the literature related to the optimization algorithm of BPNN and explain their respective advantages and shortcomings, which are obviously missing in the paper.

2.     In addition, it is suggested to add the algorithm of [1] in the introduction and experiment, which is also a novel BPNN-based method to overcome the GPS outages for INS/GPS system.

[1] G. Wang, X. Xu, Y. Yao and J. Tong, "A Novel BPNN-Based Method to Overcome the GPS Outages for INS/GPS System," in IEEE Access, vol. 7, pp. 82134-82143, 2019, doi: 10.1109/ACCESS.2019.2922212.

3. At present, there has been a lot of research on PSO-BP, so the algorithm in the paper is not novel, and the contribution made is just to apply it to integrated navigation error estimation. Is this worth publishing?

4. The introduction as a conclusion statement may mislead readers into believing that PSO-BPNN is a brand-new algorithm. It is recommended to reorganize the language to state the research contribution.

Comments on the Quality of English Language

The English expressions need to be improved. For example, using the words "futhermore", "moreover" and "additionally" together in lines 105 to 107 can cause confusion.

Reviewer 3 Report

Comments and Suggestions for Authors

The original achievement of the authors of the manuscript is the synthesis of a method for compensating errors in GPS/SINS systems for determining the ship's position by implementing neural network and particle swarm optimization algorithms.

Comments:

1. The manuscript is a poor presentation of a research report, not a scientific paper.

2. The content of the Abstract, in the first part suitable for Introduction, and in the second part for Conclusions.

3. Sentences from lines 119-127 are suitable as Abstract.

4. In the Introduction, formulate the thesis and its goals, which will become the chapters of the manuscript.

5. In Conclusions, do not summarize the work again, but present specific quantitative and qualitative conclusions from the research conducted.

6. In Conclusions, formulate a plan for further topics related to the manuscript.

Round 2

Reviewer 1 Report

Comments and Suggestions for Authors

The author has addressed the issues and suggestions related to my last review. I hava no other questions. However, the novelty and contributions of this work is a little low in my opinion.

Comments on the Quality of English Language

Minor editing of English language is needed.

Reviewer 2 Report

Comments and Suggestions for Authors

Accept in present form

Author Response

Thanks for your approve.